# Gene Expression Programming Model for Tribological Behavior of Novel SiC–ZrO_2_–Al Hybrid Composites

**DOI:** 10.3390/ma15238593

**Published:** 2022-12-02

**Authors:** Hossein Abbasi, Malihe Zeraati, Reza Fallah Moghaddam, Narendra Pal Singh Chauhan, Ghasem Sargazi, Ritamaria Di Lorenzo

**Affiliations:** 1Department of Computer Engineering, Tangestan Branch, Islamic Azad University, Ahram 75541, Iran; 2Department of Metallurgy and Materials Science, Faculty of Engineering, Shahid Bahonar University of Kerman, Kerman 76169-14111, Iran; 3Faculty of Engineering, University of Garmsar, Garmsar 59146-33817, Iran; 4Department of Chemistry, Faculty of Science, Bhupal Nobles’ University, Udaipur 313002, Rajasthan, India; 5Noncommunicable Diseases Research Center, Bam University of Medical Sciences, Bam 76617-71967, Iran; 6Facoltà di Medicina e Chirurgia, Università degli Studi di Napoli Federico II, 80138 Naples, Italy

**Keywords:** gene expression programming, tribological, hybrid, composite, zirconia

## Abstract

In order to improve product format quality and material flexibility, variety of application, and cost-effectiveness, SiC, ZrO_2_, and Al hybrid composites were manufactured in the research utilizing the powder metallurgy (PM) technique. A model was created to predict the tribological behavior of SiC–ZrO_2_–Al hybrid composites using statistical data analysis and gene expression programming (GEP) based on artificial intelligence. For the purpose of examining the impact of zirconia concentration, sliding distance, and applied stress on the wear behavior of hybrid composites, a comprehensive factor design of experiments was used. The developed GEP model was sufficiently robust to achieve extremely high accuracy in the prediction of the determine coefficient (R^2^), the root mean square error (RMSE), and the root relative square error (RRSE). The maximum state of the RMSE was 0.4357 for the GEP-1 (w1) model and the lowest state was 0.7591 for the GEP-4 (w1) model, while the maximum state of the RRSE was 0.4357 for the GEP-1 (w1) model and the minimum state was 0.3115 for the GEP-3 model (w1).

## 1. Introduction

Aluminum metal matrix composites (MMCs) have been popular over the last 20 years due to their prospective uses in the structural, aerospace, and automotive industries [1,2,3]. Due to their qualities such as high thermal conductivity, specific strength, low density, and low cost, MMCs are a desirable option. The final mechanical characteristics of a metal matrix composite are determined by the uniform distribution of the reinforcement particle. To achieve homogeneous distribution, many researchers have used methods such as mechanical alloying, casting, and powder metallurgy (PM). Due to its capacity to create parts with high homogeneity, uniform reinforcement distribution, and complex dimensionality, PM has recently attracted more attention among all fabrication techniques for the synthesis of MMCs. Manufacturing is more affordable as a result of this uniqueness, which nearly minimizes the cost of complex machining [4,5]. To enhance the tribological and mechanical characteristics, Al_2_O_3_, Si_3_N_4_, ZrO_2_, TiC, and SiC are frequently used to reinforce aluminum metal matrix composites [6,7,8,9,10]. Additionally, aluminum matrix composites reinforced with a combination of micro- and nano-sized reinforcement have garnered a lot of research interest due to their enhanced mechanical and tribological properties, making them a superior alternative to single-reinforced composites [11]. Hybrid composites with micro- and nano-size reinforcement are crucial when used as frictional materials. Due to the Orowan strengthening mechanism, nanoparticles improve the performance of the matrix and a micro reinforcement reduces the friction load, improving the hardness and tribological properties of the hybrid composites.

Because SiC enhances the tribological and mechanical properties, it has recently been widely used as a ceramic reinforcement [12,13,14]. Al/SiC composites could be improved, but at the expense of ductility and fracture toughness. Key material properties required to avoid catastrophic failure in service include ductility and the toughness of fractures [15]. Additionally, as reported in earlier studies, metal oxide shifts SiC fragility and increases Al–SiC fracture toughness when used as a reinforcement for hybrid composites [16,17]. Metal oxides can be used as a strengthening material in composites because they are affordable, widely accessible, and environmentally friendly [16]. ZrO_2_ is a readily available, reasonably priced metal oxide with exceptional thermal, mechanical, and tribological properties [6]. There are no specific results for SiC–ZrO_2_–Al hybrid composites made utilizing the PM process, according to the extensive examination of several research publications [18,19].

In the recent past, various researchers have used the GEP to estimate the various mechanical characteristics of various types of concrete [20,21]. In recent years, researchers have increasingly used the artificial neural network (ANN) technique to simulate the mechanical and tribological characteristics of composite materials [22]. The capacity to learn from small amounts of information is a very useful tool for discovering the behavior of the experimental trend much more quickly than other methods. The relationship between intricate data patterns of input and output may also be developed and predicted using the ANN technique [23,24,25]. A hydraulic model was also used to predict how well the ultrafiltration membrane would perform [26]. Regression models such as multiple linear regression (MLR) and multiple linear equation regression (MLnER) as well as optimization models like gene expression programming (GEP) have been used in recent years to predict the outcomes of complex problems and have shown to be effective and potent tools in prediction problems [27,28,29,30]. The performance of linear and nonlinear models in predicting outputs should be contrasted since the nonlinear model outperforms the linear model in small datasets [31,32]. While using GEP and statistical techniques, no comprehensive results on the tribological behavior of ZrO_2_ reinforced Al–SiC hybrid nano-composites have been found. In the current study, the tribological behavior of hybrid SiC–ZrO_2_–Al composites made by powder metallurgy was examined. The effects of the ZrO_2_ concentration, the sliding distance, and the applied load on the wear behavior of the hybrid composite were examined using a full factorial design of experiments.

## 2. Experimental and Method

### 2.1. Fabrication of Composites

Zirconia powders, silicon carbide powders, and aluminum powders all had typical particle sizes of 10 m and 30–50 nm, respectively. In the most recent investigation, these reinforcing particles were applied. The PM technique was used to create the composites. First, a quantity of elemental powder with a minimum count of 0.0001 g was weighed using a digital scale (PRECISA, Dietikon, Switzerland, ES 25SM-DR). The powders were fully combined in a centrifugal type ball mill (Fritsch, Germany) using 8 mm diameter stainless steel balls and a ball-to-powder weight ratio of 10:1 in order to accomplish homogenization and minimize particle agglomeration. The milling time and speed settings were 15 min and 100 revolutions per minute, respectively. Green compacts measuring 8 mm in diameter and 13 mm in height were created by compressing the powder combination at a pressure of 585 MPa in a uniaxial hydraulic pallet press (Type KE, Sr. No. 1327, Kimaya Engineers, Thane, India). The die wall was manually greased with zinc stearate before each compaction procedure. The compacted samples were sintered in a tube furnace at 450–470 °C for 60 min while under an argon environment (flow rate = 1.0 L·min^−1^) in order to prevent oxidation of the aluminum matrix [33]. The sintered samples were allowed to warm up to room temperature in the furnace. The densities of the samples were calculated using Archimedes’ principle [4].

### 2.2. Gene Expression Programming

The “tree-based” concept of genetic programming was originally put out by Cramer [34]. Genetic algorithms are a subclass of evolutionary algorithms (EA) that employ approaches inspired by natural evolution to develop solutions to optimization problems. The primary proponent of genetic programming, Koza [35], eventually considerably broadened genetic programming as a pioneer for a variety of difficult optimization and research problems [36]. Similar to genetic programming and genetic algorithms, gene expression programming (GEP) uses a natural selection approach (GP). It uses genetic and fitness-based population generation as well as individual selection. The foundation of GEP is individual encoding using a chromosome, a symbolic string of fixed or changeable letters [37]. The numerous genes on each GEP chromosome are each encoded by a separate sub-expression tree. Each gene structure has a head and a tail that are connected to other GEP processes. The head contains terminals (constants) and mathematical functions, while the tail just has terminals. Various functions (n) and the head (h) may be used to compute the length of the tail (t), and the result is stated as: t=h (n−1)+1.

The flowchart of the GEP algorithm can be seen in Figure 1.

## 3. Results and Discussion

The first simulation includes a critical step that determines the process parameters influencing one input layer with the zirconia concentration, sliding speed, sliding distance, and applied stress as the input and output nodes, namely wear loss. The operating characteristics as shown in Table 1 support this. Twenty-four datasets from past publications [7,38] are included in the data utilized in this study. A typical method of displaying the distribution and outliers of the input data is the box plot (Figure 2). As a result, all of the input parameters do not include any outliers, and the data generally exhibit a symmetric distribution.

On GEP models, many tests are conducted and each has a distinct setup (chromosomes number, head length, gene number and linking function, etc.). A total of candidate GEP models with a respectable level of fitness are selected out of them.

The parameters and a list of the function set utilized in the four GEP models are shown in Table 2 and Table 3, respectively.

One model is preferable to another according to a set of standards for assessing the model correctness. The root-mean-square-error (RMSE) and the root relative square error are two examples of these criteria (RRSE). To assess the consistency of the numerical values of the squared regression, a further parameter is utilized, in which the anticipated and experimental data may be calculated (R^2^). The errors and regression value are given by the following equation:(1)$$\mathrm{RMSE}=\frac{1}{\sqrt{M}}\sqrt{\overset{M}{\underset{1}{\sum }}{\left(f_{j}-f_{i}\right)}^{2}}$$
(2)$$RRSE=\frac{\sqrt{\sum_{i}{\left(f_{i}-f_{j}\right)}^{2}}}{\sqrt{\sum_{i}{\left(f_{i}-\left(\frac{1}{M}\right)\sum_{i}f_{i}\right)}^{2}}}$$
(3)$$R^{2}=1-\frac{\sum_{i}{\left(f_{i}-f_{j}\right)}^{2}}{\sum_{i}{\left(f_{j}\right)}^{2}}$$

The values of *f_i_* and *f_j_* are the actual and projected values, respectively, and *M* is the total number of datasets. R^2^, RMSE, and *RRSE* are a few examples of statistical verification criteria that have been used to assess the correctness of the fitness. Higher R^2^ and lower RMSE and RRSE values are the goals of the evaluation in order to demonstrate a more accurate methodology. Table 4 contains a summary of the statistics for the four GEP models.

The R^2^ for the four GEP models in the training and testing stages of component w1 (Figure 3) was more than 0.9740 according to the statistical findings in Table 4. The GEP-3 (w1) model showed the greatest R^2^ values for the model’s training and testing stages of 0.9740 and 0.9522, respectively. The correlations between training and testing are often not statistically different from one another in GEP models.

Figure 3 displays the R^2^ values for each of the six GEP (w1)-based models (a). The R^2^ in the maximum state (shown in this Figure for a training mode) was associated with the GEP-3 (w1) model, however, the R^2^ in the lowest state (shown in this figure for a training mode) was related to the GEP-1 (w1) model. The wear loss of hybrid composite SiC–ZrO_2_–Al was predicted by all six GEP models, although model GEP-3 (w1) is preferred over other models. The lowest RMSE was shown in Figure 3b for the training mode, where the maximum state of the RMSE was 0.4357 for the GEP-1 (w1) model and the minimum state was 0.3115 for the GEP-3 (w1) models. For the testing mode, the maximum state was 1.0906 for the GEP-1 (w1) model and the minimum state was 0.7591 for the GEP-4 (w1) model. As a result, in the maximum state for the GEP-1 (w1) model in the training mode and the lowest state for the GEP-3 (w1) model in the testing mode, respectively, the RRSE was 0.2342 and 0.1674, respectively, as shown in Figure 3c.

An evolutionary run with more chromosomes would take longer since the number of settings is often governed by the complexity of the issue and the number of potential solutions. Selecting the ideal setup parameters (such as gene head size, gene count, and linking function) for the ideal GEP model is a crucial next step.

Change the number of genes, the number of gene heads, and the linking function in all models to achieve this. The optimum GEP model structure, according to Table 2, is a GEP-3 (w4) model with a head size of 8, three genes, and a multiplication-based linking function.

The GEP-3 (w4) formulae for predicting the wear loss of the SiC–ZrO_2_–Al hybrid composites are shown in equation:
(4)$$\begin{array}{ll} \mathrm{Wear}\;\mathrm{loss}\;=\sin & \left(\log(((-5.05\times -4.08\times (\;\mathrm{Sliding}\;\mathrm{distance}\;+\;\mathrm{Applied}\;\mathrm{load}\;)\right)\\ & +(-1.55\times \;\mathrm{Sliding}\;\mathrm{distance}\;\times \;\mathrm{Concentration}\;))))\\ & \times \sin(\sin((((\;\mathrm{Sliding}\;\mathrm{distance}\;\times \;\mathrm{Applied}\;\mathrm{load}\;\times 2.54)\times \;\mathrm{Sliding}\;\mathrm{distance}\;\times 4.78)+\left(\frac{1.0}{-9.01})))\right)\\ & \times (\sin((((8.74\times \;\mathrm{Applied}\;\mathrm{load}\;\times 0.94)+(\;\mathrm{Concentration}\;\times \;\mathrm{Concentration}\;\times 8.74))\\ & +(3.87\times \;\mathrm{Sliding}\;\mathrm{distance}\;\times \;\mathrm{Concentration}\;)))+9.18) \end{array}$$


Figure 4 also includes an expression tree (ET) for the GEP-3 (w4) model.

By considering the SiC–ZrO_2_–Al hybrid composites, concentration of zirconia, sliding speed, sliding distance, and applied load as practical parameters of wear loss, a sensitivity analysis was employed for the determination of the most effective practical parameter on the microhardness of the nanocomposite coatings. Given that the best model for predicting microhardness is GEP-3 (w4), GEP-3 was employed for the sensitivity analysis (w4). In the GEP-3 (w4) models, the value for each practical parameter was set to zero while the values for the other parameters varied. By turning off the effect of the most affected parameter on the GEP-3 (w4) model, namely the RMSE as the threshold, it is reasonable to have a larger deviation from the performance of the models [24].

The current density significantly affected the wear loss of the SiC–ZrO_2_–Al hybrid composites, as can be seen in Figure 5. The influence of each process parameter on the wear attributes of the constructed composites was then examined using the created model. To ascertain the impact of zirconia concentration on wear loss, the GEP-3 (w4) simulation was run with data altering the concentration of ZrO_2_ from 5, 10, and 20% while maintaining the other process parameters such as sliding distance and applied stress, unchanged. Additionally, Figure 5 explores and illustrates the impact of other process factors.

## 4. Conclusions

Powder metallurgy was used to create the SiC–ZrO_2_–Al hybrid composites. The tribological behavior of the Al–SiC–ZrO_2_ hybrid composites was predicted using gene expression programming (GEP) and statistical data analysis for potential automotive engine-based applications. The influence of ZrO_2_ concentration, applied stress, the wear behavior of hybrid composites, and sliding distance were examined for potential engineering applications using a complete factorial design of experiments. The GEP-1 (w1) model had the highest possible RMSE of 0.4357, while the GEP-4 (w1) model had the lowest possible RMSE of 0.7591. The GEP-1 (w1) model had the highest possible RRSE of 0.4357, while the GEP-3 model had the lowest possible RRSE of 0.3115. (w1).

## Figures and Tables

**Figure 1 materials-15-08593-f001:**
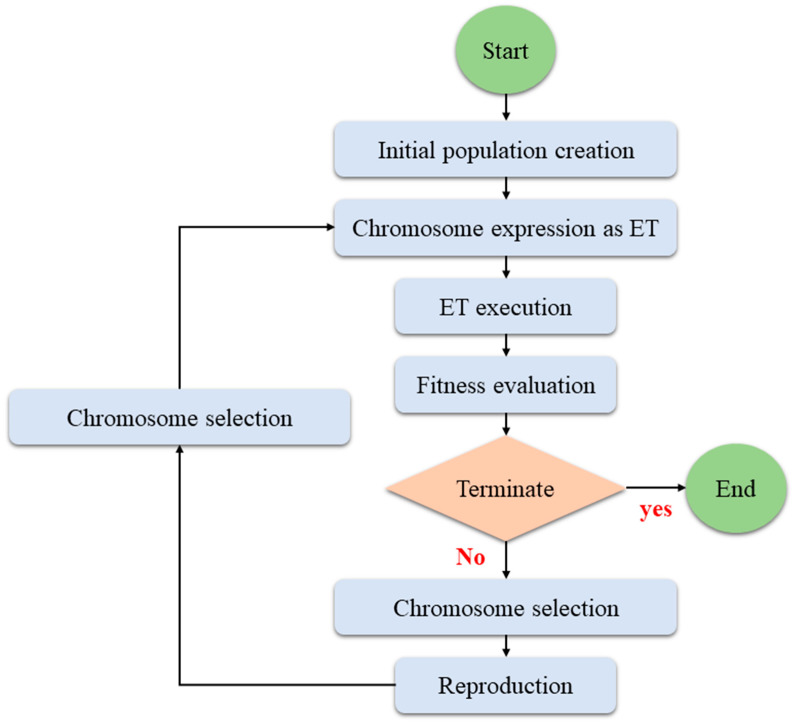
A flowchart of the GEP steps.

**Figure 2 materials-15-08593-f002:**
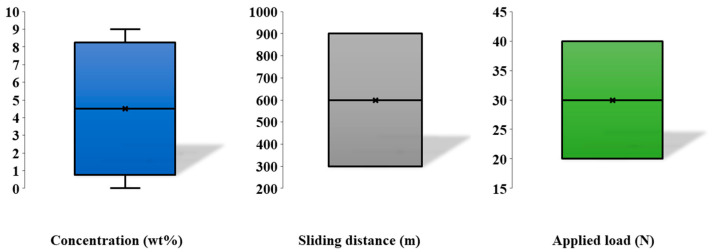
Box plot of the experimental data.

**Figure 3 materials-15-08593-f003:**
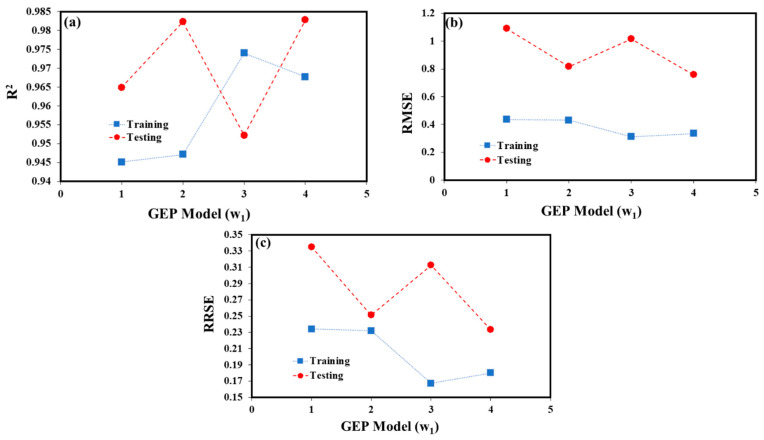
The comparison of the validation GEP (w1) model criteria (**a**) R^2^, (**b**) RMSE, and (**c**) RRSE values for the training and testing datasets.

**Figure 4 materials-15-08593-f004:**
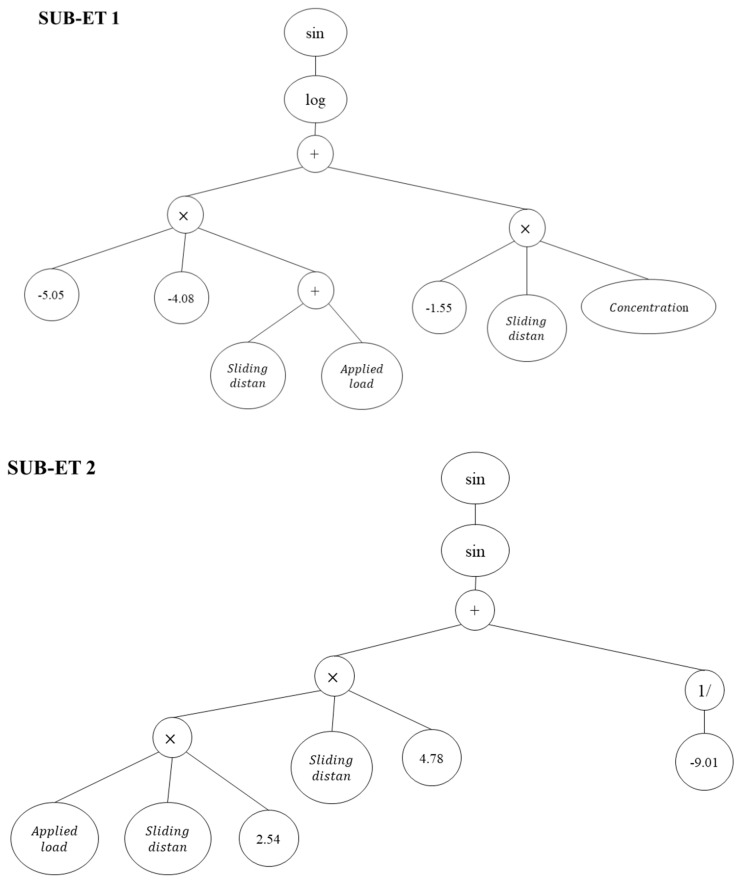
Expression tree (ET) of the GEP-3 (w4) model.

**Figure 5 materials-15-08593-f005:**
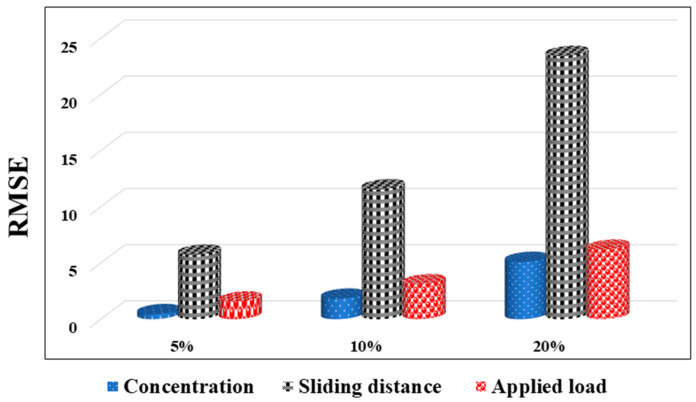
Sensitivity analysis of the practical parameters.

**Table 1 materials-15-08593-t001:** General full factorial experimental.

Concentration (wt%)	Sliding Distance (m)	Applied Load (N)
0, 3, 6, 9	300, 600, 900	20, 40

**Table 2 materials-15-08593-t002:** GEP model parameters.

**Chromosomes number**	**30**
**Head size**	7.8
**Genes number**	3, 4
**Linking function**	Addition (+), Multiplication (×)
**Fitness function error type**	RMSE
**Constant per gene**	1
**Mutation rate**	0.044
**Inversion rate**	0.1
**One-point recombination rate**	0.3
**Two-point recombination rate**	0.3
**Gene recombination rate**	0.1

**Table 3 materials-15-08593-t003:** List of function sets.

Code	Function Set
**S1**	+, ×, /, −,
**S2**	+, ×, x^2^, Sqrt
**S3**	+, sin, log, 1/x
**S4**	−, Sqrt, log, 1/x, x^2^, sin, ×

**Table 4 materials-15-08593-t004:** The statistical performances of the GEP models in RMSE, RRSE, and R^2^.

No.		Head Size	Number of Genes	Linking Function	Function Set	Training			Testing		
						R^2^	RMSE	RRSE	R^2^	RMSE	RRSE
**GEP-1**	w_1_	7	3	+	S1	0.9451	0.4357	0.2342	0.9649	1.0906	0.3353
	w_2_	7	3	×	S1	0.9677	0.3343	0.1797	0.9828	0.8902	0.2737
	w_3_	8	3	+	S1	0.9485	0.4218	0.2267	0.9436	1.1465	0.3525
	w_4_	8	3	×	S1	0.9690	0.3274	0.1760	0.9962	0.4007	0.1233
	w_5_	7	4	+	S1	0.9747	0.2955	0.1589	0.9973	0.3563	0.1095
	w_6_	7	4	×	S1	0.9762	0.2873	0.1544	0.9908	0.6120	0.1882
**GEP-2**	w_1_	7	3	+	S2	0.9471	0.4318	0.2321	0.9823	0.8175	0.2513
	w_2_	7	3	×	S2	0.9517	0.4091	0.2199	0.9757	0.8067	0.2480
	w_3_	8	3	+	S2	0.9658	0.3612	0.1942	0.9801	0.7394	0.2273
	w_4_	8	3	×	S2	0.9372	0.5564	0.2991	0.9792	0.8205	0.2552
	w_5_	7	4	+	S2	0.9425	0.4468	0.2402	0.9750	0.7548	0.2321
	w_6_	7	4	×	S2	0.9584	0.3795	0.2040	0.9798	0.6948	0.2136
**GEP-3**	w_1_	7	3	+	S3	0.9740	0.3115	0.1674	0.9522	1.0163	0.3125
	w_2_	7	3	×	S3	0.9703	0.3221	0.1731	0.9931	0.4003	0.1231
	w_3_	8	3	+	S3	0.9685	0.3462	0.1861	0.9607	1.009	0.3103
	* **w** * * _ **4** _ *	* **8** *	* **3** *	×	* **S3** *	* **0.9840** *	* **0.2357** *	* **0.1267** *	* **0.9864** *	* **0.7831** *	* **0.2408** *
	w_5_	7	4	+	S3	0.9703	0.3296	0.1772	0.9758	1.0485	0.3224
	w_6_	7	4	×	S3	0.9768	0.2847	0.1531	0.9831	0.6950	0.2137
**GEP-** **4**	w_1_	7	3	+	S4	0.9677	0.3348	0.1800	0.9828	0.7591	0.2334
	w_2_	7	3	×	S4	0.9480	0.4367	0.2347	0.9633	0.7668	0.2358
	w_3_	8	3	+	S4	0.9422	0.5103	0.2743	0.9580	1.2254	0.3768
	w_4_	8	3	×	S4	0.9538	0.4005	0.2153	0.9794	0.6767	0.2081
	w_5_	7	4	+	S4	0.9608	0.4438	0.2386	0.9750	1.1326	0.3482
	w_6_	7	4	×	S4	0.9314	0.4890	0.2629	0.9895	0.4853	0.1492

## Data Availability

Not applicable.

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
