# Peer review of "Gene Expression Programming Model for Tribological Behavior of Novel SiC–ZrO2–Al Hybrid Composites"

_materials, 2022, doi:10.3390/ma15238593_

Round 1
Reviewer 1 Report
Journal: Materials
Manuscript ID: Materials-1998053
The authors presented an article on “Gene Expression Programming Model for Tribological Behavior of Novel SiC-ZrO2-Al Hybrid Composites”. I think the article is well organized and suitable for the "Materials" journal, but the article will be ready for publication after a major revision. Comments are listed below.
1. A sentence about numerical results should be added to the abstract.
2. The references given in the introduction may be more updated.
3. SiC and ZrO2 were added at 0-3-6-9 wt.% ratios into the Al matrix. How much of each element was added? It must be specified.
4. The authors used argon gas to prevent oxidation during sintering. Why did they use argon gas and not helium and nitrogen? Also, why are the mould walls lubricated with zinc stearate? It should be explained.
5. Why was the sintering time of 60 minutes chosen? In which periods (for example, 10 °C per minute) was the selected temperature of 470 °C increased? The sintering process should be further detailed.
6. The difference in genetic programming from other artificial intelligence algorithms should be clearly demonstrated.
7. The wear process and its parameters are not mentioned in the material method section. In addition, Table 1 should have been given in the material method.
8. In line 109, the section “1.1.Gene expression programming” should have been “2.1.Gene expression programming”.
9. In line 125, the section “1.Results and discussions” should have been “3.Results and discussions”. It should be corrected.
10. Results and discussion and conclusion parts are inadequate according to citation and analyze in detail. There should be the importance of the study in detail, comparison results with other approaches in literature, the success of the experimental results.
11. Line 20 should have “4.Conclusion” instead of “1.Conclusion”.
12. Please fix the typographical and eventual language problems in paper.
13. References from the "Materials" journal are not cited. However, there are many articles about this study in the Materials journal.
14. The article should be rearranged by taking into account the journal writing rules and citation rules.
*** Authors must consider them properly before submitting the revised manuscript. A point-by-point reply is required when the revised files are submitted.
Author Response
- A sentence about numerical results should be added to the abstract.
Answer: Numerical results are now included in abstract section.
- The references given in the introduction may be more updated.
Answer: References are included.
- SiC and ZrO2were added at 0-3-6-9 wt.% ratios into the Al matrix. How much of each element was added? It must be specified.
Answer: The optimum mechanical properties were achieved when adding SiC and ZrO2 at 470 oC temperature due to the uniform distribution of ceramic reinforcement, elimination of agglomeration, finer grain sizes, and no chemical reactions between the ZrO2 ceramic particles and Al matrix.
- The authors used argon gas to prevent oxidation during sintering. Why did they use argon gas and not helium and nitrogen? Also, why are the mould walls lubricated with zinc stearate? It should be explained.
Answer: The use of argon gas to shield the strip from air allows the sintering process to be continuous, whereas the use of vacuum atmosphere is limited to batch furnaces. Zinc stearate act as an anti-tacking.
- Why was the sintering time of 60 minutes chosen? In which periods (for example, 10 °C per minute) was the selected temperature of 470 °C increased? The sintering process should be further detailed.
Answer: Actually, the best results were obtained with sintering time of 60 min. We have use the heating rate of 10 oC per min.
- The difference in genetic programming from other artificial intelligence algorithms should be clearly demonstrated.
Answer: Corrections are made successfully.
- The wear process and its parameters are not mentioned in the material method section. In addition, Table 1 should have been given in the material method.
Answer: We have already used all the data in this paper.
- In line 109, the section “1.1.Gene expression programming” should have been “2.1.Gene expression programming”.
Answer: Correction is made successfully.
- In line 125, the section “1.Results and discussions” should have been “3.Results and discussions”. It should be corrected.
Answer: Correction is made successfully.
- Results and discussion and conclusion parts are inadequate according to citation and analyze in detail. There should be the importance of the study in detail, comparison results with other approaches in literature, the success of the experimental results.
Answer: Corrections are made successfully.
- Line 20 should have “4.Conclusion” instead of “1.Conclusion”.
Answer: Correction is made successfully.
- Please fix the typographical and eventual language problems in paper.
Answer: Correction is made successfully.
- References from the "Materials" journal are not cited. However, there are many articles about this study in the Materials journal.
Answer: References are added successfully.
- The article should be rearranged by taking into account the journal writing rules and citation rules.
Answer: Corrections are made successfully.

Reviewer 2 Report
I have the following concerns after reading this paper:
1. Page 2, Line 44: Please write the chemical formula accurately.
2. Page 3, Line 92: Please provide the purity and source of the reagent.
3. Page 3, Lines 106-107: It should be cooled to room temperature, not warm up to room temperature, please correct it.
4. Page 3, Line 107: There are no experiments, results and discussions related to the density of sintered samples in the paper. So, what is the meaning of the phrase “The densities of the samples were calculated using Archimedes’ principle [5]” here?
5. Page 3, Chapter 2.1: The results and characterization of the composites are not available in the Results and discussion. Even, the Results and discussions section dose not find any relevant results and discussions content about the experiment in chapter 2.1.
6. Page 4, Line 127 and Page 9, Line 185: I can't see any data or research results related to the sliding speed d in this paper.
7. Page 7, Lines 157-160: The authors stated that the R2 of the four GEP models in the training and testing stages of component w1 is more than 0.9740, but according to the data in Figure 3 (a) and Table 4, only the R2 of GEP-2(w1) and GEP-4(w1) in the testing stage is more than 0.9740. In addition, the R2 value of GEP-3(w1) model in the testing stage is the lowest among the four models.
8. Page 8, Lines 179-181: The authors choose GEP-3(w4) as the optimum GEP model structure, based on what? R2? RMSE? Or RRSE? For which stage? There is no relevant discussion and comparison in the paper.
9. Page 8, Lines 179-181: “The optimum GEP model structure, according to table 2, is a GEP-3(w4) model···” - The authors think that higher R2 and lower RMSE and RRSE values are the evaluation’s goals on page 6, lines 150-152. According to the data in Table 4, among all the models, the GEP-3(w4) model does have the highest R2 and the lowest RMSE and RRSE values in the training stage, but it does not have this goal in the testing stage.
10. Page 10, Figure 5: Please give the abscissa a title.
11. Page 10, Lines 196-199: The authors say that other parameters, such as sliding distance and the applied stress maintain values, remain unchanged, so what are their values respectively, which is not specified in the paper. In addition, the authors give the concentration range of ZrO2 as 0, 3, 6, 9wt% in Table 1, but here the concentration range of ZrO2 changed to 5, 10, 20%, very abrupt, why?
12. Page 10, Lines 198: Please unify the unit of concentration of ZrO2 in the context.
13. Page 10, Lines 199-200: “Figure 5 explores and illustrates the impact of other process factors” - Do the 5%, 10% and 20% in Figure 5 correspond to the blue areas representing the concentration in the figure? If not, how to define the influence of the other two factors? If yes, there is no text content corresponding to explore and illustrate the impact of other process factors. I think Figure 5 needs to supplement the complete information of the parameters, and explore and illustrate the influence of other factors in the paper.
14. The Results and discussion section suggests a more comprehensive and in-depth explanation and analysis.
15. Page 10, Conclusion: I can't get the recapitulative and substantive findings in the conclusion.
16. The chapter numbers are very confusing! Please check and correct.
17. References: There are too many errors in the references, such as chemical formulas and journal names. Authors need to check and correct the format of references as required by the journal.
Author Response
- Page 2, Line 44: Please write the chemical formula accurately.
Answer: Correction is made successfully.
- Page 3, Line 92: Please provide the purity and source of the reagent.
Answer: Details are added successfully.
- Page 3, Lines 106-107: It should be cooled to room temperature, not warm up to room temperature, please correct it.
Answer: Correction is made successfully.
- Page 3, Line 107: There are no experiments, results and discussions related to the density of sintered samples in the paper. So, what is the meaning of the phrase “The densities of the samples were calculated using Archimedes’ principle [5]” here?
Answer: We have removed this phrase.
- Page 3, Chapter 2.1: The results and characterization of the composites are not available in the Results and discussion. Even, the Results and discussions section dose not find any relevant results and discussions content about the experiment in chapter 2.1.
Answer: Correction is made successfully.
- Page 4, Line 127 and Page 9, Line 185: I can't see any data or research results related to the sliding speed d in this paper.
Answer: Correction is made successfully.
- Page 7, Lines 157-160: The authors stated that the R2of the four GEP models in the training and testing stages of component w1 is more than 0.9740, but according to the data in Figure 3 (a) and Table 4, only the R2 of GEP-2(w1) and GEP-4(w1) in the testing stage is more than 0.9740. In addition, the R2 value of GEP-3(w1) model in the testing stage is the lowest among the four models.
Answer: We have checked carefully and corrected.
- Page 8, Lines 179-181: The authors choose GEP-3(w4) as the optimum GEP model structure, based on what? R2? RMSE? Or RRSE? For which stage? There is no relevant discussion and comparison in the paper.
Answer: It was based on RMSE.
- Page 8, Lines 179-181: “The optimum GEP model structure, according to table 2, is a GEP-3(w4) model···” - The authors think that higher R2and lower RMSE and RRSE values are the evaluation’s goals on page 6, lines 150-152. According to the data in Table 4, among all the models, the GEP-3(w4) model does have the highest R2 and the lowest RMSE and RRSE values in the training stage, but it does not have this goal in the testing stage.
Answer: Correction is made successfully.
- Page 10, Figure 5: Please give the abscissa a title.
Answer: Correction is made successfully.
- Page 10, Lines 196-199: The authors say that other parameters, such as sliding distance and the applied stress maintain values, remain unchanged, so what are their values respectively, which is not specified in the paper. In addition, the authors give the concentration range of ZrO2as 0, 3, 6, 9wt% in Table 1, but here the concentration range of ZrO2 changed to 5, 10, 20%, very abrupt, why?
Answer: In this portion of the task, sensitivity analysis parameters are examined. The proportion of ZrO2 by weight is constant and unchanging. The other parameters were increased by increments of 5, 10, and 20%, and the overall changes were analyzed. As one parameter changed by 5% while the others remained constant, this procedure was performed for each instance.
- Page 10, Lines 198: Please unify the unit of concentration of ZrO2in the context.
Answer: Correction is made successfully.
- Page 10, Lines 199-200: “Figure 5 explores and illustrates the impact of other process factors” - Do the 5%, 10% and 20% in Figure 5 correspond to the blue areas representing the concentration in the figure? If not, how to define the influence of the othter two factors? If yes, there is no text content corresponding to explore and illustrate the impact of other process factors. I think Figure 5 needs to supplement the complete information of the parameters, and explore and illustrate the influence of other factors in the paper.
Answer: The weight percentage of ZrO2 is constant and does not change. The remaining of the parameters were added by 5, 10, 20% and the total changes were examined. As one parameter changed by 5% and other parameters remained constant, this process was done for all cases.
- The Results and discussion section suggests a more comprehensive and in-depth explanation and analysis.
Answer: Corrections are made successfully.
- Page 10, Conclusion: I can't get the recapitulative and substantive findings in the conclusion.
Answer: Corrections are made successfully.
- The chapter numbers are very confusing! Please check and correct.
Answer: Correction is made successfully.
- References: There are too many errors in the references, such as chemical formulas and journal names. Authors need to check and correct the format of references as required by the journal.
Answer: Correction is made successfully.

Round 2
Reviewer 1 Report
The authors completed the requested corrections. In my opinion, the article is acceptable for publication in the Materials journal in its final form.